# Methyl Canthin-6-one-2-carboxylate Inhibits the Activation of the NLRP3 Inflammasome in Synovial Macrophages by Upregulating Nrf2 Expression

**DOI:** 10.3390/cimb47010038

**Published:** 2025-01-09

**Authors:** Yuanyuan Chen, Zongying Zhang, Yuan Yao, Xiaorong Zhou, Yong Ling, Liming Mao, Zhifeng Gu

**Affiliations:** 1Graduate School, Dalian Medical University, Dalian 116044, China; cyynt@ntu.edu.cn; 2Department of Rheumatology, Affiliated Hospital of Nantong University, Nantong 226019, China; 3Department of Immunology, School of Medicine, Nantong University, 19 Qixiu Road, Nantong 226019, China; 2113310013@stmail.ntu.edu.cn (Z.Z.); 2231310020@stmail.ntu.edu.cn (Y.Y.); zhouxiaorong@ntu.edu.cn (X.Z.); 4School of Pharmacy and Jiangsu Province Key Laboratory for Inflammation and Molecular Drug Target, Nantong University, Nantong 226001, China; lyyy111@sina.com; 5Basic Medical Research Center, School of Medicine, Nantong University, Nantong 226019, China

**Keywords:** methyl canthin-6-one-2-carboxylate, inflammation, rheumatoid arthritis, NLPR3, inflammasome, Nrf2, macrophage

## Abstract

Rheumatoid arthritis (RA) is an autoimmune disorder that leads to severe cartilage deterioration and synovial impairment in the joints. Previous studies have indicated that the aberrant activation of the NLRP3 inflammasome in synovial macrophages plays a significant role in the pathogenesis of RA and has been regarded as a therapeutic target for the disease. In this study, we synthesized a novel canthin-6-one alkaloid, namely methyl canthin-6-one-2-carboxylate (Cant), and assessed its effects on NLRP3 inflammasome activation in macrophages. Our data reveal that exposure to Cant significantly suppressed the transcription and secretion of multiple pro-inflammatory mediators, including IL-1β, IL-6, IL-18, TNF-α, NO, and COX2, in a dose-dependent manner. These alterations were associated with changes in the activation of various signaling pathways, including NF-kB, MAPK, and PI3K-AKT pathways. Notably, pretreatment with Cant significantly reduced LPS/ATP-induced activation of the NLRP3 inflammasome, as evidenced by the decline in the cleaved forms of IL-1β and caspase-1 in cell culture supernatants of BMDMs. Regarding the mechanisms, our data show that Cant could enhance the expression of Nrf2 in macrophages, which play an inhibitory role in ROS production. Collectively, our data demonstrate that Cant might suppress the activation of the NLRP3 inflammasome by upregulating the production of Nrf2, suggesting that Cant could serve as a candidate for the further development of anti-RA drugs.

## 1. Introduction

Rheumatoid arthritis (RA) is a chronic autoimmune disorder mainly characterized by impairments to the joints and surrounding tissues. A recent epidemiological investigation disclosed that the burden of RA varies globally and the prevalence of the disease is higher in industrialized countries, likely due to diverse genetic factors, demographics, and exposure to various environmental risk factors [1]. Despite the extensive research carried out on the pathogenesis of RA, its precise etiology remains elusive [2]. The progression of RA comprises a series of sequential phases [3]. The period before the onset of clinical joint symptoms is termed the preclinical RA phase, which itself can be subdivided into several stages [4]. In the initial stage, in a genetically predisposed individual, an aberrant immune response is triggered by host or microbial antigens arising from genetic mutations or mucosal dysbiosis outside the joint, such as in the lung and colon. During this stage, persistent immune activation leads to loss of tolerance and the production of autoantibodies, including rheumatoid factors, anti-citrullinated protein antibodies, and others. In the subsequent stage, as the autoantibody repertoire expands, local autoimmunity progresses to a systemic level. Concurrently, the levels of circulating inflammatory cytokines, chemokines, and acute phase reactants increase. In the third stage, the autoantibodies target the synovium and the bone within the joint. Patients in this stage may gradually experience joint pain and early bone loss. Environmental triggers, such as trauma or infection, can cause activated immune cells to infiltrate the synovial tissues, leading to clinically detectable synovitis [5,6].

The principal treatment modalities for RA include surgical intervention, joint stress reduction, physical and occupational therapy, and pharmacological therapies [7,8]. With advances in early diagnostic techniques and a wider range of treatment options, including the development of a variety of disease-modifying anti-rheumatic drugs (DMARDs) and biologic therapies, the treatment and long-term prognosis of RA have improved dramatically [9,10,11,12]. Among these modalities, pharmacological treatments are prioritized as the initial approach to managing RA. Many medicines have been proven to be effective in clinical applications. However, several studies have reported varying degrees of side effects from these medicines [13,14,15]. Thus, the discovery of new drugs with high efficiencies and low cytotoxicity for RA treatment constitutes an essential part of the work in drug development for treating the disease. Macrophages are key effector cells in synovial tissues and play an essential role in maintaining the homeostasis of the joint [16]. Activation of synovial macrophages may exacerbate joint inflammation by producing pro-inflammatory mediators, such as TNF-α. While specific subsets of synovial macrophages can actively regulate the remission of RA by generating lipid molecule resolvin D1 and triggering repair responses in the joint [17]. The aberrant activation of synovial macrophages has been observed in the joints of RA patients. The status of synovial macrophages has been considered to be a biomarker for the treatment response of RA patients [18]. A recent study proposed that regulating macrophage function could be a therapeutic approach for RA in an animal model of the disease [19].

The NLRP3 inflammasome is an essential component of the innate immune system and plays a role in regulating inflammatory progression in the joint [20,21,22]. It has been reported that the expression and activation of NLRP3-inflammasome-associated genes were significantly increased in the synovial tissues of RA patients [22,23,24,25]. Some studies also showed that NLRP3-inflammasome-associated genes were highly expressed in many cell types, including monocytes, macrophages, dendritic cells, endothelial cells, and B cells, while these genes were undetectable in fibroblast-like synoviocytes [24,26]. Inhibition of NLRP3 inflammasome activation has also been proven to be effective in inhibiting the inflammatory responses of monocytes and in ameliorating joint inflammation in RA patients in a collagen-induced arthritis model [24]. Thus, the NLRP3 inflammasome is involved in the pathogenesis of RA and can be a therapeutic target of the disease. The activation of the NLRP3 inflammasome can be regulated by many factors. Previous studies have shown that the antioxidant NRF2 is an important regulator of the NLRP3 inflammasome. A study by Liu et al. reported that NRF2 can inhibit NLRP3 inflammasome activation by suppressing ROS-mediated NLRP3 priming [27]. The upregulation of NRF2 correlates with a decline in NLRP3 expression and activation. The levels of Nrf2 were found to be increased in both the serum and synovial tissues of RA patients, and the levels were correlated with disease activity [28,29]. Meanwhile, the knocking-down of Nrf2 could exacerbate the proliferation and invasion of FLSs of RA patients by activating the JNK signaling pathway [28]. In comparison, activation of Nrf2 can improve arthritis in SKG mice, which carry a mutation in ZAP70 and develop RA-like joint inflammation in response to systemic exposure to β-glucan [30]. Thus, Nrf2 has been considered to be a therapeutic target of RA [31].

Methyl canthin-6-one-2-carboxylate (Cant; chemical structure shown in Figure 1) is a novel indole alkaloid derivative of canthin-6-one characterized in recent studies of ours [32,33]. We found that exposure to Cant could suppress the migration and invasion of fibroblast-like synovial cells (FLSs) in RA patients by regulating the activity of the Hippo signaling pathway [33]. Here, we aimed to explore the impact of Cant on synovial macrophages, and found that treatment with Cant could significantly inhibit the production of pro-inflammatory cytokines. Regarding the mechanisms, we showed that Cant played an inhibitory role in the activation of the NLRP3 inflammasome in macrophages, an effect associated with the upregulation of NRF2. Our study offers additional evidence, suggesting that Cant may serve as a promising candidate for the development of anti-RA therapeutics.

## 2. Materials and Methods

### 2.1. Materials and Reagents

Methyl canthin-6-one-2-carboxylate (Cant) was generated according to the literature [32]. Lipopolysaccharide (LPS, tlrl-b5lps, InvivoGen, Santiago, CA, USA) and adenosine triphosphate (ATP, tlrl-atpl, InvivoGen, Santiago, CA, USA) were purchased from InvivoGen.

### 2.2. Cell Culture

Bone-marrow-derived macrophages (BMDMs) were isolated from 8-week-old male C57BL/6J mouse femurs and tibiae as previously described [34]. In short, bone marrow cells were isolated from the femur and the tibia of mice and were then cultured in IMDM medium (L610KJ, BasalMedia, Shanghai, China) containing 10% FBS (10270-106, Gibco, Grand Island, NY, USA), 1% penicillin–streptomycin (HyClone, Logan, UT, USA), and 10 ng/mL M-CSF (315-02, PeproTech, Rocky Hill, NJ, USA) at 37 °C in a humid environment containing 5% CO_2_ for 7 days. The mouse RAW 264.7 cells were purchased from Hefei Wanwu Biotechnology Co., Ltd. (Tings-12733, Hefei, China) and were cultured in RPMI 1640 medium containing 10% FBS (FBS-S500, Newzerum, Christchurch, New Zealand) and 1% penicillin–streptomycin (HyClone, Logan, UT, USA). The procedures for animal care and experimentation in this study were approved by the Institutional Animal Care and Use Committee of Nantong University (Approval No. S20240316—800).

### 2.3. Network Pharmacology

The SwissTargetPrediction (http://swisstargetprediction.ch/ (accessed on 26 April 2024)) [35] and SuperPred (https://prediction.charite.de/ (accessed on 26 April 2024)) [36] databases were sequentially searched for Cant-related targets. A thorough search was conducted on the OMIM (https://omim.org/ (accessed on 26 April 2024)) [37], Digenet (https://www.disgenet.org/ (accessed on 26 April 2024)) [38], and Genecards (https://www.genecards.org/ (accessed on 26 April 2024)) [39] databases to identify *Homo sapiens* targets linked with RA. The distinct targets were left after removing duplicates and taking values with a correlation greater than the average. The Venn diagram was drawn using the R package ‘Venn’. The String database was utilized to conduct a protein interaction analysis of intersecting targets. The interaction factor was set to 0.4. After hiding free targets, Cytoscape 3.10.2 [40] was utilized to open the “TSV” file to construct a PPI network. The exporting network information was a “TSV” file. Network graph-related data were obtained through topology analysis. The Weisheng Xin Platform was used to perform enrichment analysis on the major nodes of RA and Cant, including Disease Ontology (DO), Kyoto Encyclopedia of Genes and Genomes (KEGG), and Gene Ontology (GO). The data were visually represented using bar charts or bubble charts.

### 2.4. CCK8 Assay

Cell viability was assessed using a cell-counting kit (CCK8 kit, abs50003—5 mL, Absin, Shanghai, China) under the same culture conditions as before. All the cells were divided into four groups: a blank group (medium only), a control group (cells + medium), and two groups with varying drug concentrations (cells + medium + 20/40 μM Cant). The cells were enumerated and adjusted to a concentration of 2 × 10^4^ cells/mL and were then planted at 100 μL/well in 96-well plates. The cells for each treatment group were seeded in three duplicate wells. The cells were cultured for the required period of time in a cell incubator at 37 °C and 5% CO_2_. Ten microliters of CCK8 solution were added to every well. The cell culture plates were left in the incubator for one hour. Lastly, the absorbance at 450 nm was measured using a plate reader. As control wells, untreated cells, medium, and CCK8 solution were also examined.

### 2.5. Western Blotting

Prior to being lysed in a 1% NP40 solution, the treated cells were washed with PBS at room temperature. The protein concentrations of the lysed samples were measured, and their concentrations were evenly adjusted. The protein samples were then heated for 10 min at 99 °C using a thermostatic heater prior to being loaded in a 4–15% SDS–polyacrylamide gel (P0466M, Beyotime, Shanghai, China). After gel running, the semi-dry transfer method (Bio-Rad Laboratories GmbH, München, Germany) was used to transfer proteins to NC membranes (Absin, China). NC membranes were then blocked in 5% skim milk powder (P0216, Beyotime, Shanghai, China) for two hours at room temperature. Membranes were incubated in solutions containing primary antibodies recognizing iNOS (#13120, Cell Signaling Technology, Boston, MA, USA), IL-1β (#31202, Cell Signaling Technology, Boston, MA, USA), COX-2 (#33345, SAB, College Park, MD, USA), p65 (#8242, Cell Signaling Technology, Boston, MA, USA), P-p65 (#3033, Cell Signaling Technology, Boston, MA, USA), p38 (#8690, Cell Signaling Technology, Boston, MA, USA), P-p38 (#4511, Cell Signaling Technology, Boston, MA, USA), ERK (#4695, Cell Signaling Technology, Boston, MA, USA), P-ERK (#4370, Cell Signaling Technology, Boston, MA, USA), JNK (#9252, Cell Signaling Technology, Boston, MA, USA), P-JNK (#4668, Cell Signaling Technology, Boston, MA, USA), AKT(#4691, Cell Signaling Technology, Boston, MA, USA), p-AKT(#4060, Cell Signaling Technology, Boston, MA, USA), PI3K(#4249, Cell Signaling Technology, Boston, MA, USA), p-PI3K (#4228, Cell Signaling Technology, Boston, MA, USA), NLRP3 (AG-20B-0014-C100, AdipoGen, Santiago, MN, USA), NRF2 (#12721, Cell Signaling Technology, Boston, MA, USA), and GAPDH (AF0006, Beyotime, Shanghai, China), respectively. After incubating at 4 °C overnight, the NC membranes were incubated in solutions containing horseradish peroxidase (HRP)-conjugated secondary antibodies (Beyotime, Shanghai, China) at room temperature for 1 h. Signal intensity was detected using WesternBright^TM^ Sirius (K-12043-D10, Menlo Park, CA, USA) using the Tanon gel imager.

### 2.6. ELISA

After treatment with Cant for 1 h, the cells were then stimulated with LPS (1 μg/mL) for 6 h. The levels of IL-6 (550950, BD Biosciences, San Jose, CA, USA) and TNF-α (abs520010, Absin, Shanghai, China) in the culture supernatants were determined using commercial ELISA kits following the manufacturer’s instructions.

### 2.7. Quantitative Polymerase Chain Reaction (qPCR)

The RNeasy Mini Kit (74104, Qiagen, Germantown, MD, USA) was utilized to extract the total cellular RNA of RA-FLS cells. The RevertAid First Strand cDNA Synthesis kit (K1622, Thermofisher, Waltham, MA, USA) was used to create complementary DNA (cDNA) samples. Following the directions on the SYBR Green RT-PCR reaction kit (A25742, Thermofisher, Waltham, MA, USA), cDNA was then amplified. The primers used for qRT-PCR analysis are listed in Table 1. The relative quantitative analysis of 2^−ΔΔCT^ was used to calculate the test findings. Every test was run three times.

### 2.8. Detection of ROS Generation

ROS levels were assessed using 2′,7′dichlorofluorescein diacetate (DCFH-DA, ab113851, abcam, San Diego, CA, USA) staining. Briefly, cells were collected and stained with 10 μm DCFH-DA for 30 min. Flow cytometry analysis was performed using a Beckman flow cytometer.

### 2.9. Statistical Analysis

All statistical analyses in the study were conducted using GraphPad Prism 8.0 software (La Jolla, CA, USA). The differences between multiple groups were analyzed using ANOVA. A significance threshold of *p* < 0.05 was applied to determine statistical significance.

## 3. Results

### 3.1. Network Pharmacological Analysis Reveals a Possible Role of Cant in Inflammatory Responses

Before exploring the effect of Cant in experiments, we carried out network pharmacological analysis to assess whether Cant has a potential effect on RA-associated inflammatory responses. A total of 202 target genes of Cant were retrieved from The SwissTargetPrediction and SuperPred database (Appendix A). Similarly, the OMIM, Disgenet, and Genecards databases were searched with “Rheumatoid arthritis” as the keyword, resulting in a total of 7032 RA target genes, excluding duplicates. By employing the R package ‘Venn’, we intersected the target genes of Cant with RA-associated genes, obtaining 75 target genes after eliminating duplicates and considering values with a correlation greater than the average (Figure 2A and Appendix A). Subsequently, a Cant–RA–target network was drawn using Cytoscape 3.10.2 software (Figure 2B). Moreover, the PPI network was established to visualize the interactions between the targets. As shown in Figure 2C, the PPI network contained 75 nodes and 517 edges. We then performed GO pathway enrichment analysis and identified 1802 statistically significant GO keywords, including 1549 BPs, 82 CCs, and 171 MFs. In Figure 2D, we show the top 10 significantly enriched GO terms for BPs, CCs, and MFs. To further acquire an understanding of the mechanisms of Cant in the inhibition of inflammation, we conducted KEGG pathway analysis and discovered that numerous signaling pathways were significantly enriched (Figure 2E). These analyses imply that Cant might have a potential function in RA-associated inflammatory responses by targeting multiple signaling pathways, such as the NF-kB and PI3K-AKT signaling pathways.

### 3.2. The Effect of Cant on Cell Viability of Macrophages

To assess the potential effect of Cant on RA-associated inflammatory responses in macrophages, we initially evaluated the potential cytotoxicity of Cant on the cells in our preliminary experiments. For this purpose, we treated mouse BMDM cells, a mouse macrophage cell line, with various concentrations of Cant; then, we examined the cell viability using the CKK8 assay at different time points. The results indicated that Cant had no effect on cell viability at concentrations up to 80 μM within 24 h (Figure 2F). Consequently, we employed concentrations of Cant lower than 80 μM in the subsequent experiments. Subsequently, we stimulated BMDM cells with LPS to establish a cellular model of inflammation, which is a widely adopted model in the literature for inflammatory responses in RA [41,42]. The concentration and stimulation times for LPS used in this study were selected in accordance with previous studies on investigating pro-inflammatory mediators in macrophages [43]. The findings of the CCK8 assay indicated that LPS stimulation exerted no effect on the cell viability of macrophages within 24 h when the concentration of Cant was no higher than 40 μM (Figure 2G).

### 3.3. Cant Inhibits the Production of Pro-Inflammatory Mediators by Macrophages

Our network pharmacological analysis revealed that Cant might target the NF-kB signaling pathway to exert its effect on inflammatory responses. Hence, in our initial test, we explored the role of Cant in the production of pro-inflammatory cytokines. This was carried out because their expressions were highly dependent on the activation of NF-kB. To achieve this, we pretreated BMDM cells with diverse doses of Cant and then stimulated the cells with LPS. Subsequently, we collected the cell culture supernatants and determined the levels of pro-inflammatory cytokines, such as IL-6 and TNF-α. The ELISA results indicated that the pretreatment with Cant significantly inhibited the secretion of both cytokines by the cells in a dose-dependent manner (Figure 3A). Furthermore, through Western blot studies, we discovered that LPS stimulation could raise the levels of iNOS, pro-IL-1β, and COX2; meanwhile, their levels were reduced in response to pretreatment with Cant (Figure 3B). In addition, exposure to Cant also inhibited LPS-triggered transcription of TNF-α, IL-6, IL-18, and COX-2 in a dose-dependent manner (Figure 3C). In subsequent studies, we utilized RAW264.7 cells, another mouse macrophage cell line, to validate the findings observed in BMDM cells; here, we determined that Cant could also inhibit the production of pro-inflammatory mediators by RAW264.7 cells induced by LPS stimuli (Figure 3D–F). Consequently, these data indicated that Cant exerted an inhibitory effect on LPS-induced immune responses in macrophages.

### 3.4. Cant Inhibits NF-kB and MAPK Signaling Pathways

Through further exploration, we elaborated on the mechanisms of the inhibitory effect of Cant on the production of pro-inflammatory cytokines. Here, the BMDM cells were pretreated with diverse concentrations of Cant; then, the cells were stimulated with LPS and harvested at different time points. The cell lysates were utilized in the Western blot assay for detecting various inflammation-associated signaling pathways. As shown in Figure 4A, LPS stimulation increased the level of p65 phosphorylation, an essential part and indicator of NF-kB signaling pathway. While exposure to Cant significantly reduced LPS-triggered p65 phosphorylation in BMDM cells (Figure 4A,B). Similarly, we examined the alterations of the three MAPK signaling pathways and discovered that exposure to Cant stimulation exerted significant inhibition on the phosphorylation of AKT1, p38, ERK, and JNK (Figure 4A,B). In concurrent studies, we inspected the variations in the aforementioned signaling pathways in RAW264.7 cells following exposure to Cant and witnessed comparable results to those of BMDM cells (Figure 4C,D). Additionally, we examined the impact of Cant on the activation of PI3K-AKT pathway and found that Cant could suppress the phosphorylation of AKT1, but have no effect of PI3K phosphorylation in BMDMs (Figure 4E,F). Thus, these data indicated that the inhibitory effect of Cant on pro-inflammatory mediators was mediated through its inhibition of the NF-kB, MAPK, and AKT signaling pathways.

### 3.5. Cant Suppresses NLRP3 Inflammasome Activation in Macrophages

The NLRP3 inflammasome is a crucial regulator of the inflammatory response. To explore the potential impact of Cant on the activation of the NLRP3 inflammasome, we pretreated mouse BMDM cells with diverse concentrations of Cant. Subsequently, the cells were primed with LPS (200 ng/mL) for 8 h and were stimulated with ATP, an activator of the NLRP3 inflammasome, for 30 min. We then collected the culture supernatants to determine the levels of IL-1β p17 and caspase-1 p20 by Western blots. The results indicated that LPS plus ATP stimulation elicited the release of both IL-1β p17 and caspase-1 p20 (Figure 5A), whereas pretreatment with Cant significantly suppressed the release of both molecules into the culture supernatants (Figure 5A). Meanwhile, the LPS/ATP-stimulated cells were also harvested and lysed for the detection of the expression of NLRP3 inflammasome-associated molecules. The findings revealed that LPS exposure significantly elevated the levels of NLRP3 and pro-IL-1β (Figure 5A), while ATP and Cant had a minimal effect on the expression of NLRP3, pro-IL-1β, and pro-caspase-1 (Figure 5A). Thus, these data demonstrated that Cant could suppress LPS/ATP-induced activation of the NLRP3 inflammasome in macrophages.

### 3.6. Cant Inhibits ROS Production and Enhances Nrf2 Expression in BMDMs

The activation of the NLRP3 inflammasome is closely related to the oxidative stress resulting from the accumulation of intracellular ROS. To explore whether the inhibitory role of Cant on NLRP3 inflammasome activation was associated with the alterations of cellular ROS, we determined the intracellular ROS levels using flow cytometry and discovered that LPS significantly elevated the level of ROS, while pretreatment with Cant significantly suppressed the LPS-induced production of ROS (Figure 5B). This result was consistent with the observations in the above pathway analysis, indicating that Cant target genes were significantly enriched in genes associated with ROS regulation. Our analyses also suggested that Cant might have a role in regulating the function of NFE2L2 (Figure 2B,C), a gene encoding Nrf2, which is one of the central regulators of cellular redox status. To explore whether Nrf2 plays a role in the inhibitory effect of Cant on NLRP3 inflammasome activation, we examined if the level of Nrf2 had changes during the activation of the NLRP3 inflammasome. The results demonstrated that LPS could induce the downregulation of Nrf2, while Cant exposure significantly upregulated the level of Nrf2 (Figure 5C). These data suggested that Cant may inhibit the activation of the NLRP3 inflammasome by promoting the expression of Nrf2.

## 4. Discussion

In this study, we probed into the potential role of the canthin-6-one derivative, Cant, in RA-related inflammatory responses using mouse cellular models of macrophages. The pharmacological analysis revealed that Cant might bind to numerous targets implicated in RA-associated signaling pathways. We then utilized the mouse macrophage model stimulated by LPS to assess the role of Cant in the production of pro-inflammatory cytokines. Our results indicated that pretreatment with Cant could markedly inhibit the levels of pro-inflammatory cytokines and other inflammatory mediators triggered by LPS, including iNOS, NO, IL-6, IL-1β, and TNF-α. Subsequent mechanistic analyses revealed that Cant exerted an inhibitory effect on the activation of the NLRP3 inflammasome, an outcome that might be related to the inhibitory effect of Cant on the NF-kB signaling pathways, which provides a priming signal for NLRP3 inflammasome activation. Notably, the elevated level of Nrf2 and the decreased level of cellular ROS observed in our study might constitute evidence that Cant suppresses the second signal (activating signal) of NLRP3 inflammasome activation by upregulating Nrf2.

The first query we must address prior to commencing the study of Cant in RA is whether Cant has the ability to affect RA-related inflammatory responses. We thus conducted network pharmacology analyses and discovered that Cant can target numerous genes, which intersected with many genes associated with RA progression. Subsequent analyses indicated that these genes were enriched in multiple signaling pathways implicated in inflammatory responses, including the NF-kB pathway, suggesting that Cant might play a role in regulating inflammatory responses in RA. Given that the pathological basis of RA is synovitis, it is imperative to assess the potential roles of Cant on various cells in the initiation and progression of synovitis, such as the fibroblast-like synoviocytes (FLSs) and macrophages. In our previous study, we investigated the effect of Cant on FLSs and found that exposure to Cant could suppress various functions of FLSs associated with synovitis [33]. In the present study, we focused on the effects of Cant on synovial macrophages. Ideally, primary synovial macrophages would be the best choice for this purpose. However, obtaining sufficient numbers of these cells from mice is challenging due to their limited presence in mouse synovial tissues [44]. Therefore, we selected two macrophage cell lines, RAW 264.7 cells and BMDMs, as substitutes to mimic the function of synovial macrophages, which have been widely used in previous RA studies [45,46,47,48]. We thus stimulated the cells with LPS to establish a cellular model of inflammation and found that the production of multiple pro-inflammatory mediators, including IL-1β, IL-6, TNF-α, iNOS, and NO, was reduced after exposure to Cant, suggesting an inhibitory role of Cant on inflammatory mediators. These discoveries were similar to our previous observations in FLS cells after treatment with Cant [33]. It should be noted that the primary compound of Cant, canthin-6-one, was also observed to have inhibitory effect on pro-inflammatory mediators. The study by Yue et al. [49] showed that canthin-6-one could attenuate LPS-induced release of pro-inflammatory mediators by astrocytes. Thus, the inhibition of pro-inflammatory mediators might be a common characteristic of canthin-6-one derivatives, while their relative effects on RA-related inflammatory responses might vary and should be evaluated in future studies.

The transcription of pro-inflammatory mediators, including IL-1β, was largely dependent on the NF-kB pathway [50]. Our network pharmacological analyses also offered evidence of potential bindings between Cant and NF-kB. We therefore proceeded to explore the effect of Cant on NF-kB activation. As anticipated, our data indicated that pretreatment with Cant could significantly inhibit the phosphorylation of p65, suggesting a decrease in NF-kB activation. Meanwhile, we examined the impact of Cant on MAPK signaling pathways, which are also critical pathways in inflammatory responses. Some components of MAPK pathways, such as MAPK4, were also identified as Cant targets in our analyses. Our data revealed that three MAPK pathways, including ERK, JNK, and p38, were also suppressed by Cant exposure. Our findings provide further evidence that the NF-kB and MAPK are common signaling pathways employed by numerous compounds to exert anti-inflammatory effects. However, in many cases, a specific compound may simultaneously target additional signaling pathways involved in inflammatory responses. This phenomenon may partially explain the differential effects observed among different compounds in a disease context, despite their shared influence on particular inflammatory pathways.

Following the observation of Cant’s effect on NF-kB activation, we subsequently investigated the influence of this compound on the activation of NLRP3 inflammasome, as NF-kB activation serves as a priming signal for NLRP3 inflammasome activation. As expected, our data showed that treatment with Cant markedly reduced the release of IL-1β p17 and caspase-1 p20, suggesting a decline in NLRP3 inflammasome activation. It should be noted that the inhibitory effect on NLRP3 inflammasome activation has also been observed in canthin-6-one, which has a similar effect on NLRP3 inflammasome activation in astrocytes by regulation of the NF-kB signal [49]. The effects of various compounds on the NLRP3 inflammasome have also been reported in many other studies in various arthritis models. For instance, using collagen-induced arthritis in mice, Zhang et al. [51] found that dihydroarteannuin, a derivative of artemisinin isolated from the herbal plant *Artemisia annua* L., could alleviate bone destruction and paw edema of DBA/1J mice by inhibiting the activation of NLRP3 inflammasome. Furthermore, cardamonin [52] and mastoparan M [53] could ameliorate joint inflammation by suppressing NLRP3 inflammasome activation in iron-overload-induced arthritis and MSU-crystal-induced gouty arthritis, respectively. Therefore, the NLRP3 inflammasome is a common target of many compounds in suppressing joint inflammation. NLRP3 may be not the direct binding partner of the compounds, which may bind to various upstream regulators of NLRP3 to regulate the activity of the inflammasome, and, therefore, affect the progression of arthritis.

Identifying the upstream regulator of the NLRP3 inflammasome that is targeted by Cant is therefore a crucial issue for elucidating the mechanisms through which Cant exerts its anti-inflammatory role. The activation of the NLRP3 inflammasome is closely associated with cellular redox conditions. Interestingly, in the data of our network pharmacological analyses, we noticed that Cant could target NFE2L2, which encodes Nrf2, one of the primary redox regulators in RA [54]. The knock-down of Nrf2 could exacerbate TNF-α-induced migration and invasion of RA-FLS cells by activating the JNK signaling pathway [28], suggesting that Nrf2 may play a suppressive role to joint inflammation in RA. In this study, we found that exposure to Cant could significantly enhance the level of Nrf2 in mouse macrophages. This finding suggested that Nrf2 may serve as a mechanism by which Cant suppresses the activation of the NLRP3 inflammasome. In future studies, we will investigate the question of whether Nrf2 plays a role in Cant’s anti-inflammatory function.

Our network pharmacological analyses also indicate that AKT1 is one of the core genes targeted by Cant to achieve its anti-inflammatory role. The KEGG analysis also indicated that the PI3K-AKT pathway is one of the major signaling pathways enriched by Cant targets in RA. We therefore evaluated the effect of Cant on the activation of PI3K-AKT1 and found that Cant significantly reduced the phosphorylation of AKT1, but did not affect PI3K phosphorylation in BMDMs. The role of the AKT pathway in inflammatory responses is controversial. Some previous studies have presented evidence that the PI3K-AKT pathway can suppress the activation of immune cells and the production of multiple pro-inflammatory cytokines [55,56]; meanwhile, other studies have indicated that the PI3K-AKT pathway may positively regulate joint inflammation [57,58]. For instance, a study by Anh Vo et al. reported that AKT1 was responsible for the anti-inflammatory effect of 1-p-Coumaroyl β-D-Glucoside on LPS-induced inflammation in RAW264.7 cells [59]. Moreover, the PI3K-AKT signaling pathway was also found to be important for curcumin-induced anti-inflammatory effects in microglia [55]. Additionally, ATK1 has also been reported to be a suppressor of the NLRP3 inflammasome activation by regulating NLRP3 phosphorylation of serine 5 [60]. In comparison, the study by Lu et al. reported that the PI3K/AKT/mTOR pathway was inhibited by oroxin B, and thus alleviated osteoarthritis [58]. Wang et al. reported that IL-1β-mediated inflammatory responses and cartilage damage could be suppressed by Cucurbitacin E through inhibiting the PI3K/AKT pathway in chondrocytes [57]. Therefore, the role of AKT in inflammation may be affected by the inflammatory environment. In future studies, we will conduct further tests to determine whether the reduced phosphorylation of AKT1 is involved in Cant’s anti-inflammatory function or whether it was only a concomitant alteration in its activation.

Nevertheless, our research is not without limitations. First of all, we evaluated the effect of Cant on the activation of the NLRP3 inflammasome in macrophages, an effect which may also be applicable in other cell types such as lymphocytes and dendritic cells in synovial tissues. Further studies are needed to clarify the relative impact of Cant on other cell types associated with the progression of RA. Secondly, the physiological role of Cant has not been evaluated using an in vivo model of RA in this study. We will test the effect of Cant on NLRP3 inflammasome activation in tissues of collagen-induced arthritis in future studies. Thirdly, the effect of Cant on NLRP3 inflammasome activation may also be affected by other signaling pathways. In this study, we only tested the PI3K-AKT pathway, the NF-kB pathway, and the MAPK pathway. The question of whether other signaling pathways, such as the HIF1α signaling pathway, were involved in the inhibitory role of Cant on NLRP3 inflammasome activation needs to be determined in the future. Additionally, the findings for Cant in our study were observed in RAW 264.7 cells and BMDMs. In future studies we plan to explore alternative methods to obtain sufficient primary synovial macrophages or employ advanced techniques such as single-cell sequencing to evaluate the contribution of these cells to Cant’s effect on RA.

In conclusion, our data demonstrate that Cant exerts an inhibitory effect on the activation of the NLRP3 inflammasome in macrophages. This inhibition is mediated by the upregulation of the expression of Nrf2 and the downregulation of the amount of ROS in macrophages. Our findings suggest that Cant is a promising candidate in regulating the activation of the NLRP3 inflammasome in macrophages. Prior to further development of Cant as a new anti-RA drug in clinical studies, extensive evaluation is required regarding several key aspects: the detailed mechanisms by which Cant affects Nrf2 expression and ROS accumulation in macrophages; its potential effect on multiple species of animal models of arthritis; and its potential in in vivo toxicities for various organs. Based on the cumulative evidence from these studies, a clinical trial could be conducted to systematically evaluate the efficacy and safety of Cant for patients with RA.

## Figures and Tables

**Figure 1 cimb-47-00038-f001:**
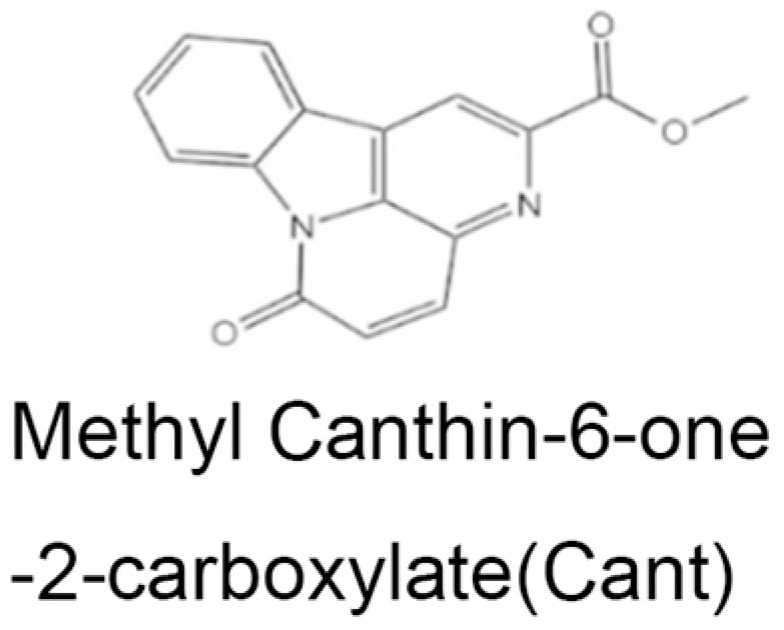
The chemical structure of methyl canthin-6-one-2-carboxylate (Cant).

**Figure 2 cimb-47-00038-f002:**
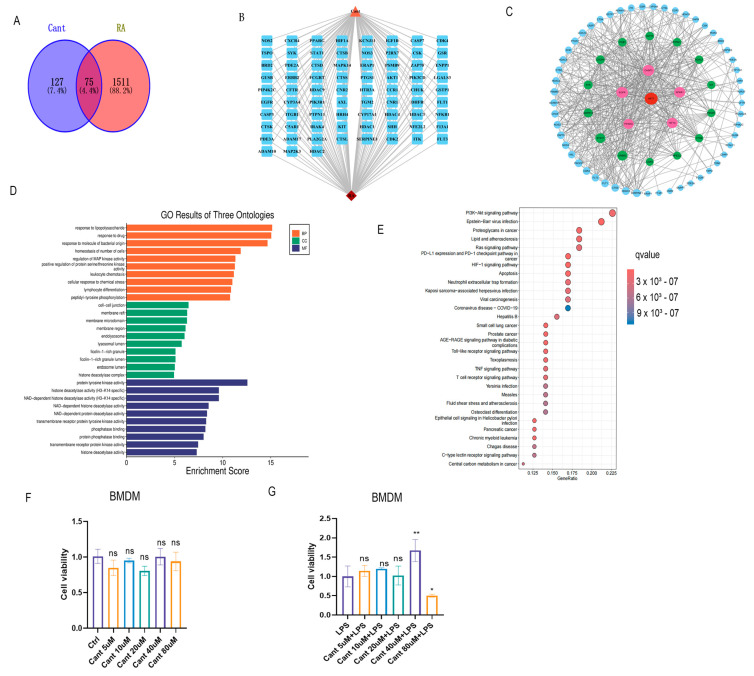
Pharmacological analysis indicates a potential anti-RA role of Cant. (**A**) Cant target genes were searched using SwissTargetPrediction and SuperPred databases and were intersected with RA-related genes. (**B**) A Cant–RA–target network was drawn using Cytoscape 3.10.2 software. (**C**) The PPI network was established to visualize the interactions between the targets. (**D**) GO pathway enrichment analysis identified the enriched terms of BPs, CCs, and MFs associated with Cant targets. (**E**) KEGG pathway analysis identified the enriched signaling pathways associated with Cant target genes. (**F**) Mouse BMDM cells were exposed to various concentrations of Cant; then, the cell viability were examined using CKK8 assay at indicated time points. (**G**) BMDM cells were exposed to various concentrations of Cant; then, the cells were stimulated with LPS, and the cell viability was detected using a CKK8 assay at indicated time points. Data in (**F**,**G**) were presented as means ± SD (*n* = 3) (* = *p* < 0.05, ** = *p* < 0.01, ns = not significant, vs. control group).

**Figure 3 cimb-47-00038-f003:**
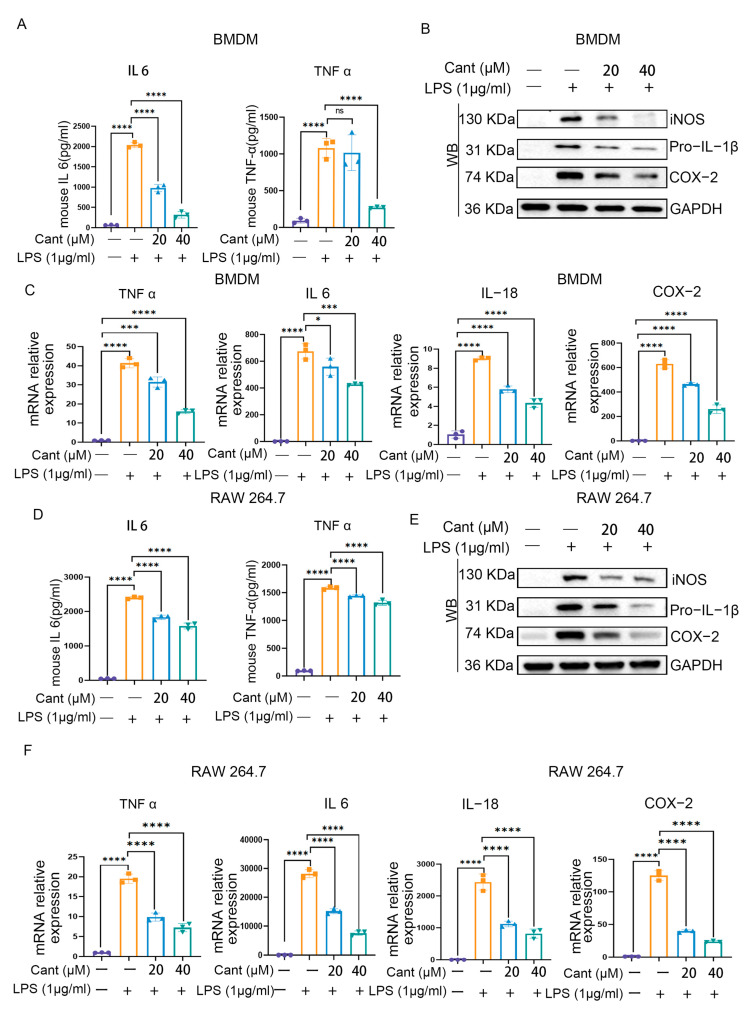
Cant inhibits the production of pro-inflammatory mediators. (**A**) Mouse BMDM cells were exposed to indicated concentrations of Cant and then were stimulated with LPS; the culture supernatants were collected for detection of IL-6 and TNF-α by ELISA. (**B**) BMDM cells were treated as indicated in (**A**); the cells were lysed and subjected to Western blot detection of iNOS, pro-IL-1β, and COX-2; GAPDH was used as an internal control. (**C**) BMDM cells were treated as indicated in (**A**); the cells were subjected to RNA extraction and subsequent detection of transcription of TNF-α, IL-6, IL-18, and COX-2 by RT-PCR. RAW264.7 cells were used to validate the results obtained using BMDMs. (**D**) IL-6 and TNF-α levels in culture supernatants of the cells were detected using ELISA. (**E**) iNOS, pro-IL-1β, and COX-2, GAPDH levels in cell lysates were detected using Western blot. (**F**) The transcription of TNF-α, IL-6, IL-18, and COX-2 was detected using RT-PCR. The data in (**A**,**C**,**D**,**F**) are presented as means ± SD (*n* = 3) (ns = not significant, *, *p* < 0.05, ***, *p* < 0.001, **** *p* < 0.0001, vs. LPS group).

**Figure 4 cimb-47-00038-f004:**
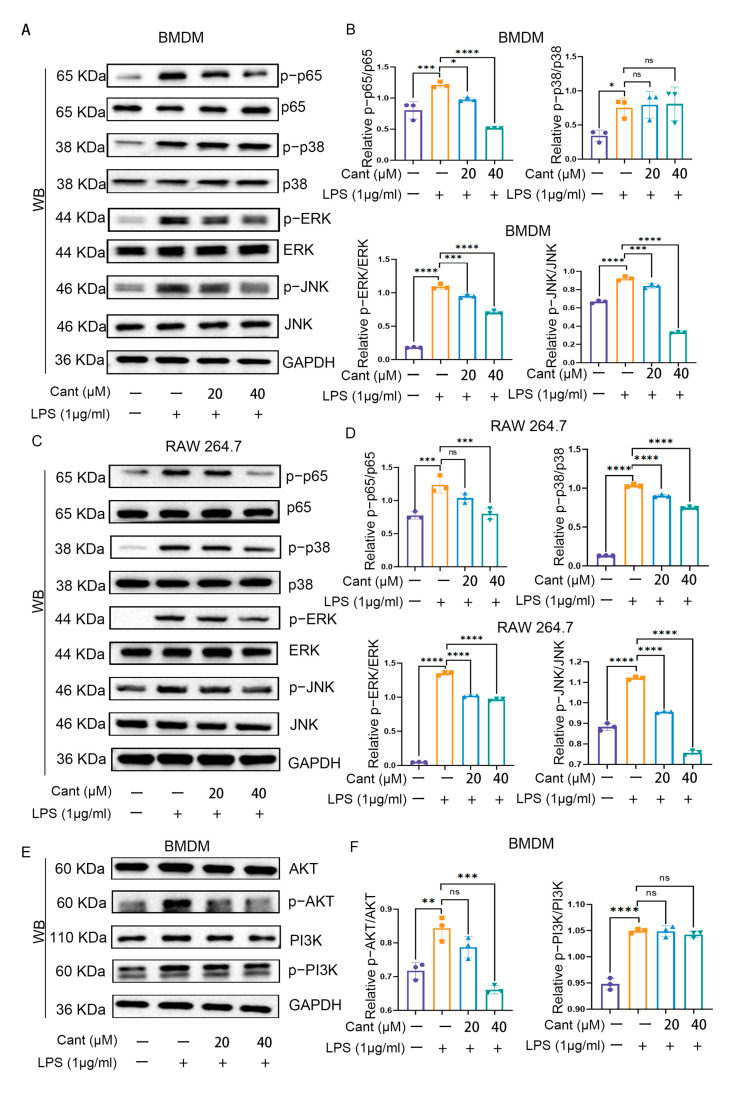
Cant inhibits the NF-kB and MAPK signaling pathways. (**A**) BMDM cells were pretreated with indicated concentrations of Cant; the cells were then stimulated with LPS for 2 h. Then, the cells were lysed, and the cell lysates were subjected to detection of phospho-p65, p65, phospho-p38, p38, phospho-ERK, ERK, phospho-JNK, and JNK using Western blot analysis. GAPDH was used as an internal control. (**B**) Quantification of relative levels of phosphorylated p65, p38, ERK, and JNK in comparison to the non-phosphorylated proteins in (**A**). The same stimulating method was applied to RAW264.7 cells, and the same proteins were detected using Western blot (**C**). The quantified data are shown in (**D**). (**E**) The BMDM lysates were subjected to detection of AKT, phospho-AKT, PI3K, and phospho-PI3K using Western blot analysis. GAPDH was used as an internal control. (**F**) Quantification of relative level of phosphorylated AKT and PI3K in comparison to the non-phosphorylated proteins in (**E**). The data in (**A**,**C**,**E**) are representative of three independent experiments. Data in (**B**,**D**,**F**) are presented as means ± SD. * = *p* < 0.05; ** = *p* < 0.01; *** = *p* < 0.001; **** = *p* < 0.0001. One-way analysis of variance was used. ns = not significant.

**Figure 5 cimb-47-00038-f005:**
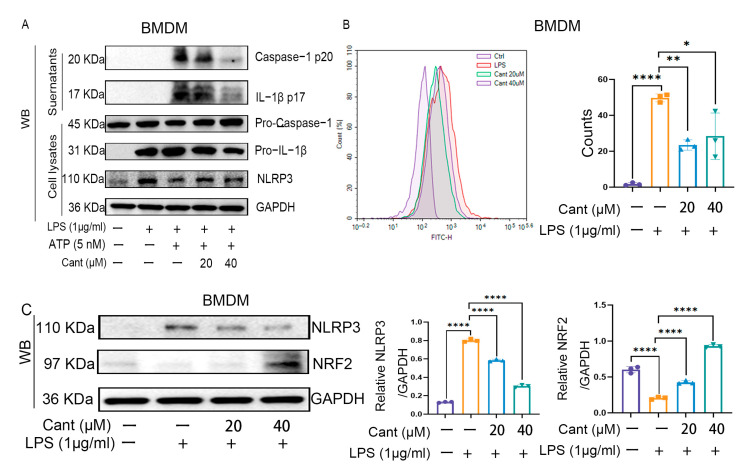
Cant suppresses the activation of the NLRP3 inflammasome by promoting the level of Nrf2 and inhibiting ROS production. BMDMs were stimulated with LPS (1 μg/mL) for 6 h in serum-free culture medium and were then treated with various concentrations of Cant. The cells were then stimulated with ATP (5 mM) for 20 min to activate the NLRP3 inflammasome. The total proteins in the cell culture supernatants were precipitated and applied in Western blot analysis of caspase-1 p20 and IL-1β p17. The cells were harvested and lysed, the cell lysates were subjected to Western blot analysis of pro-caspase-1, pro-IL-1β, and NLRP3. GAPDH was used as an internal control. (**B**) BMDMs were stimulated as indicated in (**A**) and then the cells were subjected to ROS detection using DCFH-DA staining using flow cytometry. (**C**, left) The cell lysates obtained in (**A**) were used in the detection of NLRP3 and Nrf2 using Western blot. The quantification of the expression levels of NLRP3 (**C**, middle) and Nrf2 (**C**, right) are shown. The data are representative of three independent experiments. Data in (**B**,**C**) are presented as the means ± SD. * = *p* < 0.05; ** = *p* < 0.01; **** = *p* < 0.0001. One-way analysis of variance was used.

**Table 1 cimb-47-00038-t001:** The primers used in qRT-PCR.

Gene		Sequence
*GAPDH*	F	CATCACTGCCACCCAGAAGACTG
	R	ATGCCAGTGAGCTTCCCGTTCAG
*TNF-α*	F	GGTGCCTATGTCTCAGCCTCTT
	R	GCCATAGAACTGATGAGAGGGAG
*IL-6*	F	TACCACTTCACAAGTCGGAGGC
	R	CTGCAAGTGCATCATCGTTGTTC
*IL-18*	F	GACAGCCTGTGTTCGAGGATATG
	R	TGTTCTTACAGGAGAGGGTAGAC
*COX-2*	F	GCGACATACTCAAGCAGGAGCA
	R	AGTGGTAACCGCTCAGGTGTTG

## Data Availability

The main contributions of this study are included in this article, further inquiries can be directed to the corresponding author.

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
