# Peer review of "Methyl Canthin-6-one-2-carboxylate Inhibits the Activation of the NLRP3 Inflammasome in Synovial Macrophages by Upregulating Nrf2 Expression"

_cimb, 2025, doi:10.3390/cimb47010038_

Round 1

Reviewer 1 Report

Comments and Suggestions for Authors

The present manuscript evaluates the implications of Methyl Canthin-6- Canthin-6-one-2-carboxylate in inhibition of Activation of the NLRP3 Inflammasome in Synovial Macrophages by Upregulating Nrf2 Expression. The topic is interesting but some major aspects should be solved before acceptance. Please see my concerns related to this paper:

1. The concluding part of the summary should be improved in terms of future research directions to which this research can be related.

2.  It is very important to make a clear differentiation between the inflammatory pathophysiologic mechanisms in the pre RA phase and the RA phase proper.

3. The index [x] is a structure per se and does not stick to another word (L43 factors[1]). Please check and correct throughout the manuscript.

4. Therapeutic and adjuvant management of this complex pathology should be presented. I suggest you check and consult the following updated sources: PMID: 36058148 and PMID: 33998910.

5. I suggest that Figure 1 should be split because this way it is not readable and part of it should be included in the introduction chapter and the rest in the related results area. It is not advisable that the figure should appear and its mention in the main text should be after the figure.

6. Computer programs/softwares (L111 etc.) should be included as references of the web page, not just the link.

7. I suggest that you follow the IMRAD structure in the template provided by the journal related to the order of the sections in the instructions for authors.

8. The transition of these results further into clinical management and how they could be implemented in future clinical evaluations should be emphasized.

9. A conclusion section is needed to summarize the most important aspects of the publication associated with future research directions that may address the limitations of the current study.

Author Response

The present manuscript evaluates the implications of Methyl Canthin-6- Canthin-6-one-2-carboxylate in inhibition of Activation of the NLRP3 Inflammasome in Synovial Macrophages by Upregulating Nrf2 Expression. The topic is interesting but some major aspects should be solved before acceptance. Please see my concerns related to this paper:

  1. The concluding part of the summary should be improved in terms of future research directions to which this research can be related.

Response: We appreciate the reviewer’s comment and have included several sentences in the concluding part of the summary to discuss the directions for future studies (please see in page 14, line 504-514).

  1. It is very important to make a clear differentiation between the inflammatory pathophysiologic mechanisms in the pre RA phase and the RA phase proper.

Response: We thank the reviewer’s comment and have included several sentences to clearify the inflammatory pathophysilogic mechanisms in the pre-RA and the RA phase (please see in page 2, lines 46-59).

  1. The index [x] is a structure per se and does not stick to another word (L43 factors[1]). Please check and correct throughout the manuscript.

 Response: Thanks for the reviewers' suggestion. We have made corrections at the appropriate places in the manuscript and have scrutinized the rest of the manuscript.

  1. Therapeutic and adjuvant management of this complex pathology should be presented. I suggest you check and consult the following updated sources: PMID: 36058148 and PMID: 33998910.

Response: Thanks for the reviewers' suggestion. In response to your suggestions, we have added descriptions of RA pathology treatment and adjunctive management in the appropriate places in the manuscript, with citations to the literature that you recommend (please see in page 2, line 62-66 and references 9, 11).

  1. I suggest that Figure 1 should be split because this way it is not readable and part of it should be included in the introduction chapter and the rest in the related results area. It is not advisable that the figure should appear and its mention in the main text should be after the figure.

 Response: Thanks for the reviewers' suggestion. We have revised Figure 1 in accordance with the reviewer’s suggestion. The chemical structure of Cant (original Figure 1A) has been moved to the Introduction section and is designated as Figure 1, while the remaining content of the the original Figure 1 has been relocated to the Results section and is now designated as Figure 2.

  1. Computer programs/softwares (L111 etc.) should be included as references of the web page, not just the link.

 Response: We thank the reviewers' suggestion and have included corresponding references for computer programs/softwares utilized in our analysis (please see in pages 3-4, lines 137-146).

  1. I suggest that you follow the IMRAD structure in the template provided by the journal related to the order of the sections in the instructions for authors.

 Response: We appreciate the reviewer’s question. We have structured this manuscript by referring to the IMRAD structure in the template provided by the journal in relation to the order of chapters in the author's notes.

  1. The transition of these results further into clinical management and how they could be implemented in future clinical evaluations should be emphasized.

Response: We appreciate the reviewer’s suggestion and have included several sentences to emphasize Cant’s potential roles in clinical applicaition and the related studies that should be carried out to promote this process (please see in page 14, lines 504-514).

  1. A conclusion section is needed to summarize the most important aspects of the publication associated with future research directions that may address the limitations of the current study.

Response: We thank the reviewer’s comment and have included a conclusion paragraph to summarize the major findings and directions that may address the limitations of the study (please see in page 14, lines 504-514).

Reviewer 2 Report

Comments and Suggestions for Authors

Rheumatoid arthritis (RA) is an autoimmune disease characterized by erosive arthritis, and its pathological basis is synovitis. The early symptoms of the disease include morning stiffness, swelling, and pain in the joints. Finally, joint deformities may occur and normal joint function may be lost. Authors found that cant might suppress the activation of the NLRP3 inflammasome by up-regulating the production of Nrf2, suggesting that Cant could serve as a candidate for the further development of anti-RA drugs. I think the experimental data is detailed, solid, and the conclusions are reliable, but there are some small issues that need to be addressed.

1) synovial macrophages plays a significant role in the pathogenesis of RA and has been regarded as a therapeutic target for the disease, another important issue, all experiments conducted by the authors were validated on mouse macrophages and BMDMs. How can you achieve the above goals?

2) Animal welfare certificate not displayed.

3) There are many small errors in the writing of the article, and the author should carefully proofread them.

4) More authoritative journals should be used as references, as there are many high-quality articles in NLRP3.

Comments on the Quality of English Language

Rheumatoid arthritis (RA) is an autoimmune disease characterized by erosive arthritis, and its pathological basis is synovitis. The early symptoms of the disease include morning stiffness, swelling, and pain in the joints. Finally, joint deformities may occur and normal joint function may be lost. Authors found that cant might suppress the activation of the NLRP3 inflammasome by up-regulating the production of Nrf2, suggesting that Cant could serve as a candidate for the further development of anti-RA drugs. I think the experimental data is detailed, solid, and the conclusions are reliable, but there are some small issues that need to be addressed.

1) synovial macrophages plays a significant role in the pathogenesis of RA and has been regarded as a therapeutic target for the disease, another important issue, all experiments conducted by the authors were validated on mouse macrophages and BMDMs. How can you achieve the above goals?

2) Animal welfare certificate not displayed.

3) There are many small errors in the writing of the article, and the author should carefully proofread them.

4) More authoritative journals should be used as references, as there are many high-quality articles in NLRP3.

Round 2

Reviewer 1 Report

Comments and Suggestions for Authors

The authors have significantly improved the manuscript based on the suggestions received.

Author Response

We thank the reviewer's positive comment and kind help to our work.

Reviewer 2 Report

Comments and Suggestions for Authors

The quality of the revised manuscript has significantly improved, but I still haven't received a satisfactory answer to my first question?
